# XPC deficiency increases risk of hematologic malignancies through mutator phenotype and characteristic mutational signature

Andrey A. Yurchenko [1], Ismael Padioleau [1], Bakhyt T. Matkarimov [2], Jean Soulier[3], Alain Sarasin[4] & Sergey Nikolaev [1✉]

Recent studies demonstrated a dramatically increased risk of leukemia in patients with a rare genetic disorder, Xeroderma Pigmentosum group C (XP-C), characterized by constitutive deficiency of global genome nucleotide excision repair (GG-NER). The genetic mechanisms of non-skin cancers in XP-C patients remain unexplored. In this study, we analyze a unique collection of internal XP-C tumor genomes including 6 leukemias and 2 sarcomas. We observe a specific mutational pattern and an average of 25-fold increase of mutation rates in XP-C versus sporadic leukemia which we presume leads to its elevated incidence and early appearance. We describe a strong mutational asymmetry with respect to transcription and the direction of replication in XP-C tumors suggesting association of mutagenesis with bulky purine DNA lesions of probably endogenous origin. These findings suggest existence of a balance between formation and repair of bulky DNA lesions by GG-NER in human body cells which is disrupted in XP-C patients.

[1] INSERM U981, Gustave Roussy Cancer Campus, Université Paris Saclay, Villejuif, France. [2] National Laboratory Astana, Nazarbayev University, 010000 Astana, Kazakhstan. [3] University of Paris, INSERM U944 and CNRS UMR7212, Institut de Recherche Saint-Louis, F-75010, Paris, France. [4] CNRS UMR9019 Genome Integrity and Cancers, Institut Gustave Roussy, Université Paris-Saclay, Villejuif, France. ✉email: sergey.nikolaev@gustaveroussy.fr

Xeroderma Pigmentosum (XP) is a group of rare recessive genetic disorders, which includes seven complementation groups (A−G) characterized by constitutive deficiency of Nucleotide Excision Repair (NER) pathway, and XP variant (loss of polymerase η[1]). NER serves as a primary pathway for repairing various helix-distorting DNA adducts. The NER is subdivided into global genome (GG-NER) and transcription-coupled (TC-NER) sub-pathways that preferentially operate genome-wide and on the transcribed DNA strand of genes respectively. XP patients demonstrate striking tumor-prone phenotype with near 10,000-times increased risk of non-melanoma skin cancer and 2000-times risk of melanoma due to the inability of cells to efficiently repair the major UV photoproducts[2,3]. XP complementation group C (XP-C) characterized by GG-NER deficiency (but with an unaffected TC-NER) is one of the most tumor susceptible subtypes of XP[4]. Moreover, it was hypothesized that XP patients may harbor 10−20 times increased risk to some types of internal tumors including leukemia, sarcomas[5] and thyroid nodules[6,7].

Two recent studies reported a more than a thousand-fold increased risk of hematological malignancies in independent cohorts of XP-C patients[8,9], which demonstrated mainly myelodysplastic syndrome with secondary acute myeloid leukemia manifestation. The genetic mechanism of increased risk of internal tumors in XP patients is not well understood.

Experiments with animal XP-C models demonstrated high incidence of liver and lung cancer[10] as well as 30-fold increase of spontaneous mutation rate in *Hprt* gene in T-lymphocytes of 1-year-old mice[11]. Induction of oxidative stress has been shown to further increase the somatic mutagenesis in $Xpc^{-/-}$-deficient mice with steady accumulation with age[12]. A similar tumor-prone phenotype was observed in *Ddb2/Xpe*-deficient mice with impaired GG-NER pathway: these animals developed broad spectrum of tumors with particularly high incidence of hematopoietic neoplasms[13].

In this work we perform whole-genome sequencing (WGS) of a unique collection of internal tumors from XP-C patients to demonstrate that the constitutive GG-NER deficiency causes mutator phenotype rendering susceptibility to hematological malignancies. A particular genomic mutational signature explains the majority of mutations in the studied XP-C leukemias and sarcomas. Observed mutational profiles indicate that this mutational process is associated with lesions formed from purine bases. This work explores mutational patterns and their mechanisms in XP-C patients beyond cutaneous malignancies genome-wide.

## Results

**XP-C leukemia is characterized by mutator phenotype.** We sequenced whole genomes of six myeloid leukemia, one uterine rhabdomyosarcoma and one breast sarcoma along with paired normal tissues from unrelated patients, representing XP-C, the most frequent group of XP in Northern Africa and Europe[14] and created a catalog of 202,467 somatic mutations (Table 1 and Supplementary Table 1). Seven out of eight samples harbored a founder c.1643_1644 delTG mutation characteristic of the given XP-C population[14] (Table 1). The patients developed internal tumors early in life, between 12 and 30 years of age (median age of tumor diagnosis—24 years). XP-C cancers contained somatic copy number aberrations (SCNAs) and mutations which are characteristic for corresponding types of sporadic malignancies: mutations in *TP53* and deletions of chromosomes 5 and 7 in leukemia, biallelic loss of *CDKN2A* in breast cancer and highly unstable genome of rhabdomyosarcoma (Supplementary Table 1). We compared frequency of *TP53* mutations and common chromosomal aberrations (5q and 7q deletions) between XP-C leukemia and adult de novo acute myeloid leukemia cohort[15] (AML); and found that *TP53* was mutated significantly more often in our dataset (5/6 cases and 15/200 cases for XP-C and sporadic AML respectively, $P = 2.963e−05$, odds ratio = 58.66, 95% CI = 6.04−2872.04; Fisher's exact test, two-sided). Together with significantly higher proportions of 5q and 7q deletions in XP-C leukemia ($P = 1.024e−06$ for 5q and $P = 0.002985$ for 7q deletions, Fisher's exact test, two-sided), this may indicate that the studied leukemia cases are close to *TP53* mutated with complex karyotype subgroup according to Papaemmanuil et al.[16].

We identified 14.5−31.2 (mean 24.6)-fold increase in the number of somatic mutations in XP-C leukemia samples relative to the sporadic myeloid neoplasms (Mann–Whitney U test, two-sided, $P = 5.8e−05$) and the absence of such an effect for XP-C sarcomas (Fig. 1a). This effect was consistent for single base substitutions (SBS), small indels (ID) and double base substitutions (DBS, Fig. 1a).

**XPC deficiency underlies characteristic mutational process.** The genomic mutational profiles in XP-C tumors were similar between each other irrespectively of the tumor type (average pairwise Cosine similarity of 0.964 (from 0.886 to 0.998)) (Fig. 1b, c, Supplementary Fig. 1 and Supplementary Data 1) but were different from tissue-matched sporadic tumors (Fig. 1b, c). The distinct grouping of XP-C tumors based on mutational profiles was further confirmed in the context of 190 sporadic tissue-matched cancers by multidimension scaling analysis (Fig. 1b). The mutational patterns of indels were dominated by single nucleotide deletions of C:G and T:A bases in homopolymer stretches and dinucleotide deletions in repeats (Supplementary Fig. 1b). The dinucleotide substitutions were not overrepresented by specific classes and demonstrated a broad range of contexts (Supplementary Fig. 1c).

To better understand the mutational processes operating in XP-C cancers, we extracted mutational signatures from XP-C and sporadic tissue-matched tumors with the non-negative matrix factorization approach[17] (NMF). Seven signatures were extracted from this dataset (Supplementary Fig. 2a, b) and one of them, Signature "C" explained on average 83.1% of mutations in the

**Table 1 Description of the studied XP-C tumors.**

| Sample | Geographic familial origin | Homozygous *XPC* gene mutation[a] | Diagnosis | SBS | ID | DBS |
|---|---|---|---|---|---|---|
| SA009T1 | Europe | delTG | RAEB-2, AML[b] with MDS-related changes | 36,745 | 3147 | 235 |
| SA002T2 | North Africa | delTG | Uterine Rhabdomyosarcoma[b] | 5692 | 280 | 43 |
| SA006T2 | North Africa | delTG | AML[b] | 33,230 | 2322 | 198 |
| SA008T6 | North Africa | delTG | RAEB-1, RAEB-2, AML-6[b] | 37,783 | 2377 | 245 |
| SA012T2 | North Africa | delTG | AML-6[b] | 33,821 | 2575 | 160 |
| SA007T3 | East Africa | IVS12 | Breast sarcoma[b] | 4787 | 451 | 34 |
| SA010T2 | North Africa | delTG | AML-6[b] | 17,685 | 1722 | 99 |
| SA011T2 | North Africa | delTG | T-ALL, RAEB-1, AML[b] | 17,274 | 1464 | 98 |

*RAEB* refractory anemia with excess blasts, *AML* acute myeloid leukemia, *MDS* myelodysplastic syndrome, *T-ALL* T-cell acute lymphoblastic leukemia.
[a]delTG refers to c.1643_1644 delTG; p.Val548AlafsX572 [14], IVS12 refers to the splice site mutation NM_004628:exon13:c.2251-1 G > C[34].
[b]Tumor samples used for genomic sequencing.

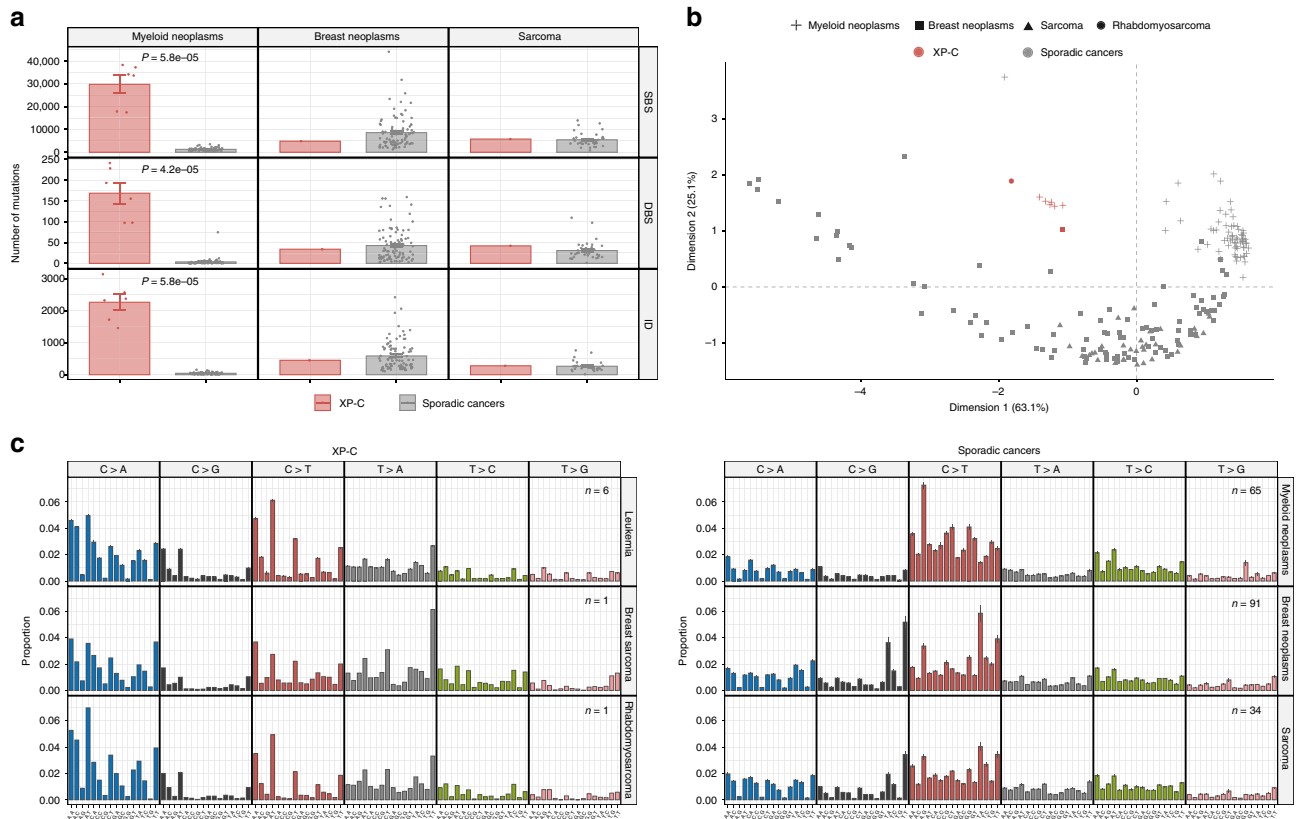

**Fig. 1 Mutational load and profiles of XP-C and 190 tissue-matched sporadic cancers. a** Number of SBS (single base substitutions), DBS (double base substitutions) and ID (indels) in XP-C and sporadic cancers with indicated SEM intervals. The difference is highly significant for myeloid neoplasms (Mann–Whitney $U$ test, two-sided, $n = 6$ for XP-C and $n = 65$ for myeloid neoplasms), but number of mutations in breast sarcoma ($n = 1$) and rhabdomyosarcoma ($n = 1$) are in the range of sporadic tumors ($n = 91$ and $n = 34$ for breast cancer and sarcoma respectively). **b** Multidimension scaling plot based on the Cosine similarity distance between the mutational profiles of the samples. XP-C tumors clearly groups together and are distant from tissue-matched sporadic cancers. **c** Trinucleotide-context mutational profiles (SEM intervals are shown in case of multiple samples, $n$ represents the number of independent cancer samples). An x-axis represents the nucleotides upstream and downstream of mutation. XP-C tumors demonstrate high similarity with each other (left panel), but profiles of sporadic cancers (right panel) are different from them.

XP-C samples (57% in breast sarcoma, 88.9% in rhabdomyosarcoma and 84.1−88.7% in leukemia) while in sporadic tumors only small contribution (average 9.7%, range 0−34.3%) of signature "C" was observed (Fig. 2a, b and Supplementary Fig. 2c, d).

These seven extracted signatures (A−G) together with original XP-C mutational profiles were compared with COSMIC mutational signatures[18] and mutational profiles of organoids from human *XPC* and mouse *Ercc1* knockouts[19] using unsupervised clustering. This analysis revealed that the XP-C tumor mutational profiles and their NMF-derived mutational Signature "C" had the highest similarity to the COSMIC Signature 8 (cosine similarity of 0.87−0.92, and 0.86 respectively) and formed a cluster together with *XPC* and *Ercc1* organoid knockouts (Fig. 2c and Supplementary Fig. 2e). At the same time the Signature "C" was different from Signature 8 by strong transcriptional asymmetry, increased mutations from C and decreased mutations from T (1.24- and 1.43-fold respectively) specifically in excess of VpCpT > D and NpCpT > T (where V designates A,C,T and D−A, G,T; Fig. 2a).

**Mutational asymmetries in XP-C tumors.** A mutational process associated with XPC deficiency is expected to demonstrate asymmetry between the transcribed and untranscribed strands of a gene[20] (transcriptional bias: TRB). This may be associated with excess of unrepaired bulky lesions on the untranscribed strand due to impaired GG-NER while on the transcribed strand such

lesions would be effectively repaired by TC-NER[21]. Indeed, transcriptional strand bias in XP-C was strong and highly significant for all six classes of nucleotide substitutions grouped by the reference and mutated nucleotide, while in tissue-matched sporadic cancers it was weak or absent (Fig. 3a−c, e and Supplementary Fig. 3a−c). Moreover, the strongest transcriptional bias was detected in highly expressed genes of XP-C tumors, reaching 7.34-fold (Wilcoxon signed-rank test, two-sided, $P = 2.91\mathrm{e}{-11}$) in XP-C leukemia (Fig. 3c and Supplementary Fig. 3d).

These effects could be explained by either excess of mutations from damaged pyrimidines or decrease of mutations from damaged purines on the transcribed (noncoding) strand. Both phenomena were previously described (see Haradhvala et al.[21]) and refer to transcription-coupled damage (TCD) or transcription-coupled repair (TCR). In case of TCD the increase of mutation rates in gene as compared to intergenic region should be observed (TCD in liver cancer analysis in Haradhvala et al.[21]) while in case of TCR we can expect the decrease of mutation rates in gene as compared to intergenic regions. In order to discriminate between these two possibilities, a comparison between mutation rates in intergenic and genic regions separately for purines and pyrimidines can be performed. To validate the suspected effect of TC-NER (decrease of mutations from purines on the transcribed strand), we performed two analysis. First, we compared relative signature contributions on the transcribed and untranscribed strands of genes and observed strong depletion of the predominant in XP-C

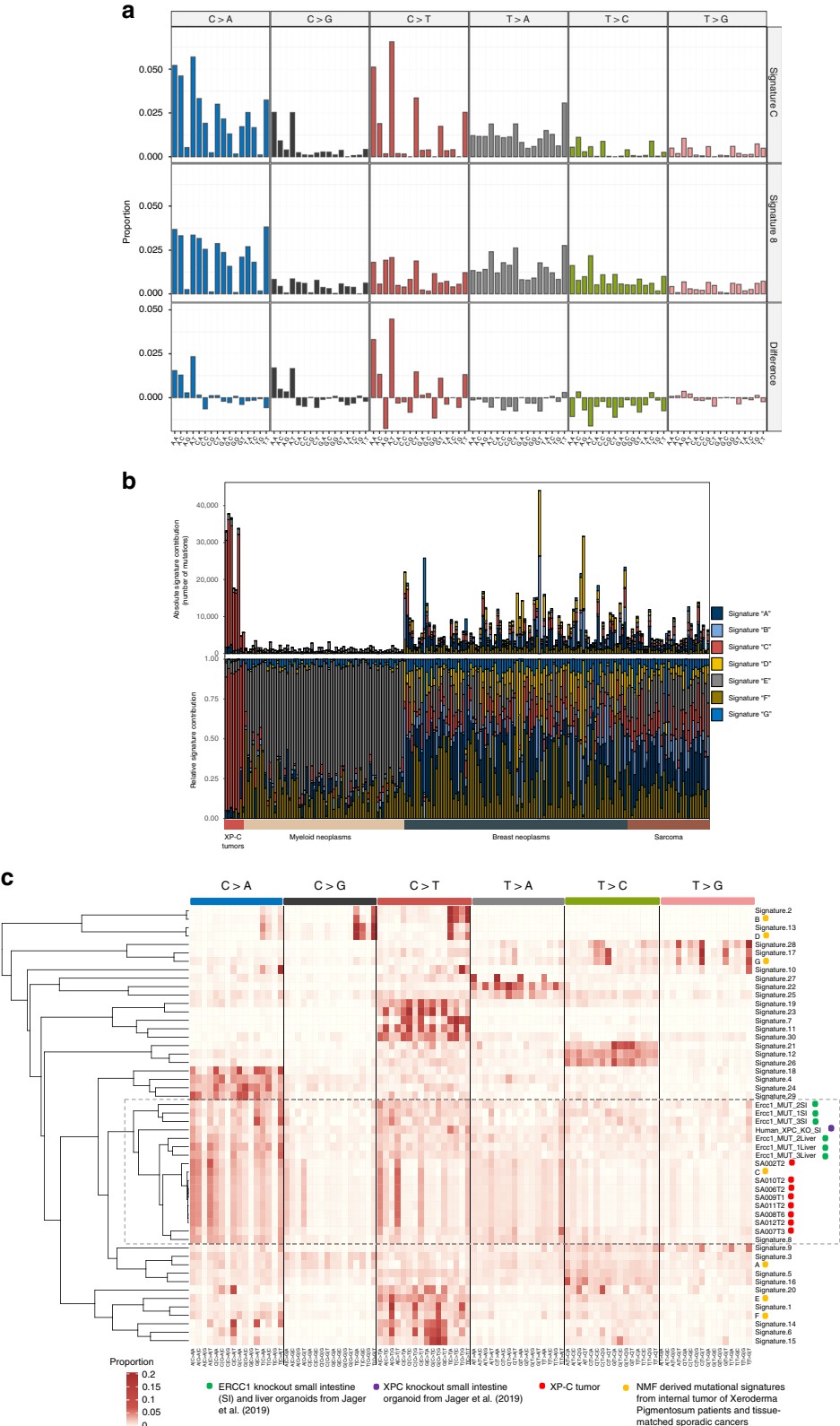

**Fig. 2 Mutational profiles of XP-C tumors in the context of known mutational signatures. a** NMF-derived mutational Signature "C" from XP-C tumors and tissue-matched sporadic cancers in comparison with COSMIC Signature 8 [18] (Cosine similarity = 0.86). **b** Relative contribution of NMF-derived mutational signatures in XP-C and tissue-matched sporadic cancers (NMF approach). XP-C tumor mutational profiles are dominated by Signature "C", while sporadic cancers by other signatures with relatively small proportion of Signature "C". **c** Unsupervised hierarchical clustering based on the Cosine similarity distances between the XP-C tumors mutational profiles, NMF-derived mutational signatures from XPC tumors and tissue-matched sporadic cancers, COSMIC mutational signatures (Signatures 1–30), and *XPC* and *Ercc1* organoid knockouts [19]. XP-C tumors cluster with each other and COSMIC Signature 8 forming a larger cluster with *Ercc1* and *XPC* organoid knockouts.

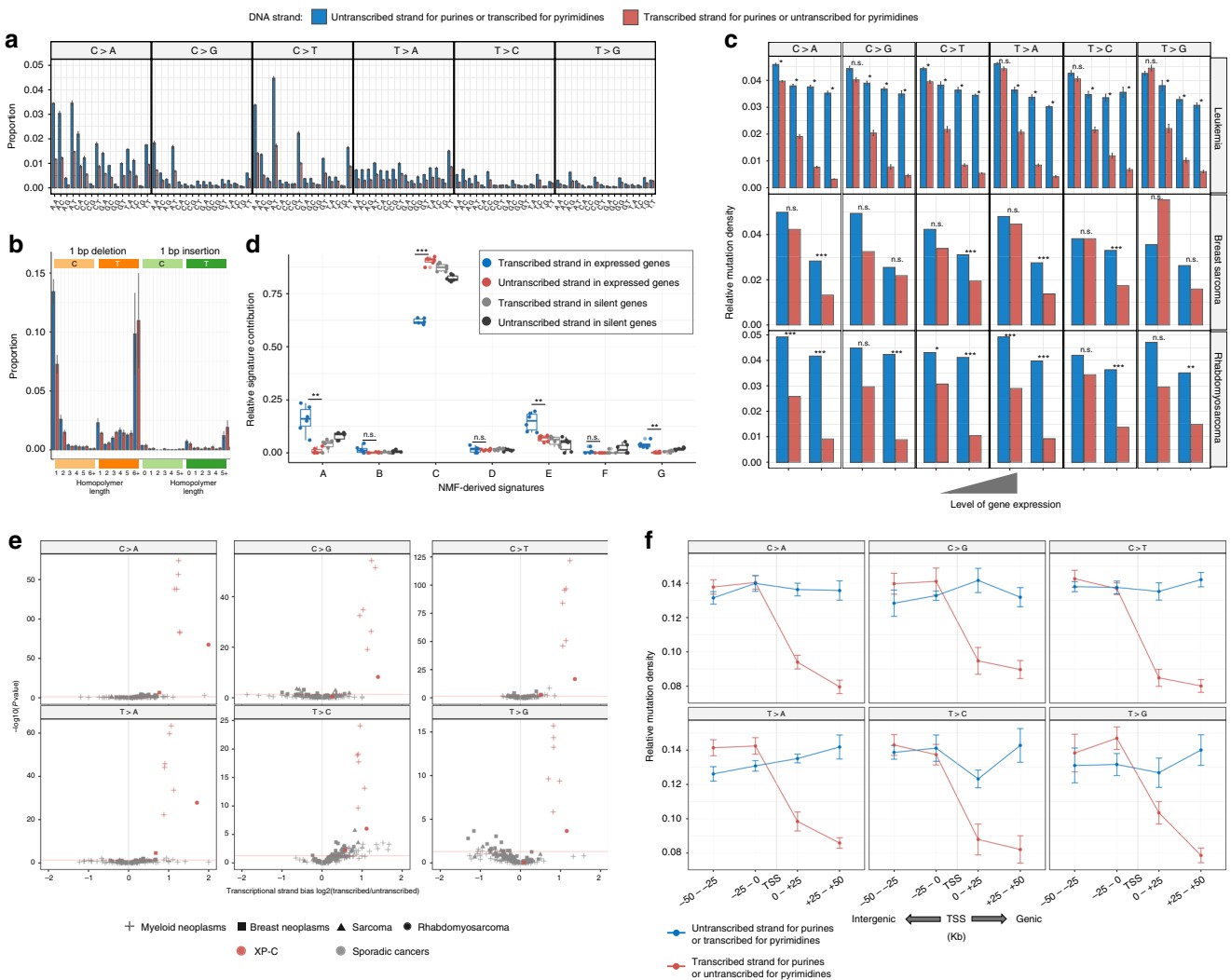

**Fig. 3 Strong transcriptional bias (TRB) is a specific feature of XP-C tumors. a** TRB is observed in the majority of trinucleotide contexts of XP-C leukemia samples ($n = 6$, SEMs are indicated). **b** TRB is highly pronounced for specific single nucleotide C:G deletions in XP-C leukemia samples ($n = 6$, SEMs are indicated). **c** TRB strength depends on the level of gene expression and is most pronounced in highly expressed genes (SEMs are indicated for leukemia; Poisson, two-sided test used for breast sarcoma ($n = 1$) and rhabdomyosarcoma ($n = 1$); Wilcoxon signed-rank, two-sided test for leukemia ($n = 6$), P: ns—nonsignificant, *<0.5, **<0.01, ***<0.001). **d** Relative mutational signature contribution for mutations separated by transcribed and untranscribed strands in transcriptionally active (FPKM > 2) and silent genes (FPKM < 0.05) of XP-C leukemia. Boxes depict the interquartile range (25–75% percentile), lines—the median, whiskers—1.5× the IQR below the first quartile and above the third quartile. Predominant in XP-C leukemia Signature "C" is depleted on the transcribed strands with functional TC-NER, but relative contribution of signatures "A" and "E" typical for sporadic leukemia is enriched on the transcribed strand (t test, two-sided, paired between transcribed and untranscribed strands in expressed genes ($n = 6$), P: ns—nonsignificant, *<0.5, **<0.01, ***<0.001). **e** TRB is highly significant and pronounced in XP-C samples for all six substitution classes in comparison with sporadic cancers (Poisson two-sided test). **f** The strong TRB observed in XP-C leukemia ($n = 6$) is caused by transcriptional-coupled repair (TC-NER) but not transcriptional-associated damage. Strong decrease of mutation rate is observed on the genic untranscribed strand for pyrimidines (transcribed for purines, red; right side of transcription start site, TSS), but not on the transcribed strand for pyrimidines (untranscribed for purines, blue) as compared to neighboring intergenic regions (±50 kbp from transcription start site, SEMs are indicated).

leukemia Signature "C" as well as increase of typical for sporadic leukemia Signatures "A" and "E" on the transcribed strand of genes (Fig. 3d). Second, we compared mutation rates separately on transcribed and untranscribed strands of genes with proximal intergenic regions and observed a strong and significant effect compatible with the decrease of mutations from purines on the transcribed strand (average 1.64-fold, Wilcoxon signed-rank test, two-sided, $P = 1.694e{-}13$) while there was no significant difference of mutations from purines between intergenic regions and untranscribed strand ($P = 0.4437$; conventional mutation representation depicts decrease of mutations from pyrimidines on the untranscribed strand; Fig. 3f and Supplementary Fig. 3e). In line

with that, we observed no difference between mutations from purines on untranscribed strand and intergenic regions at different replication times, while signature of repair of mutations from purines on transcribed strand was observed and was the strongest in early-replicating regions which are usually associated with active gene transcription[20,22] (Fig. 4a and Supplementary Fig. 4a). Similarly to SBS, transcriptional bias in DBS and ID indicated that the primary damage is on purine bases, specifically in CpC > ApD and single nucleotide deletion of C:G nucleotides (Fig. 3b and Supplementary Fig. 3b, c).

Recent report suggested that bulky DNA lesions on the lagging strand during DNA replication are more frequently converted

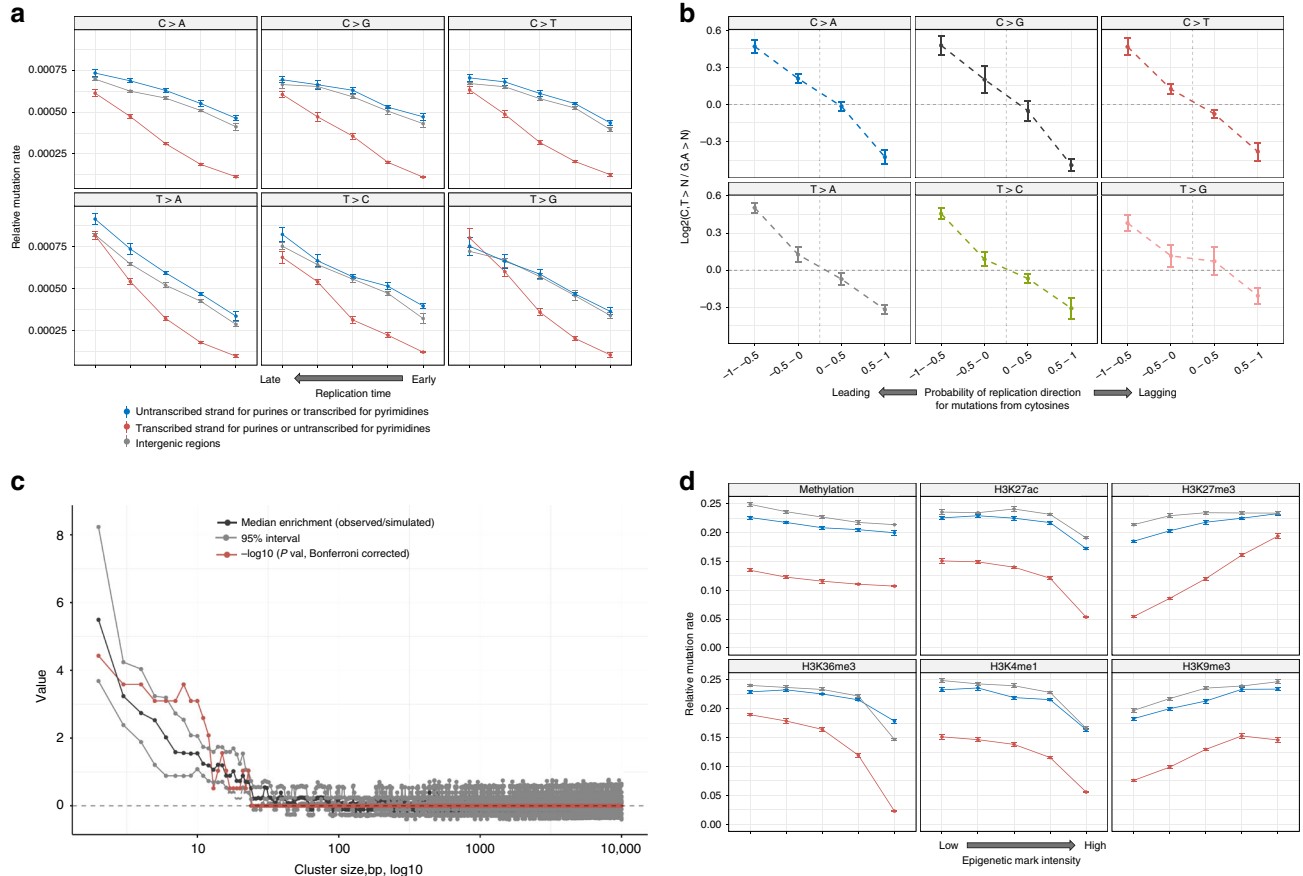

**Fig. 4 Genomic landscape of mutagenesis in XP-C internal tumors. a** Mutational density on the transcribed, untranscribed DNA strands of genes as well in intergenic regions presented as a function of replication timing in XP-C leukemia ($n = 6$, SEMs are indicated). Replication time is split onto five quantiles. Mutation rate for pyrimidines on the transcribed strand (or purines on the untranscribed, blue) is not different from intergenic regions within the same bin, which is compatible with the absence of GG-NER. **b** Pyrimidine/purine ratios of mutation rates for regions of the genome grouped by propensity of reference DNA strand to be replicated as leading (left) or lagging (right) strand during DNA synthesis for XP-C leukemia genomes ($n = 6$, SEMs are indicated). Strong enrichment of mutagenesis on the leading strand from pyrimidines (C and T) (lagging strands from purines (G and A)) is observed for all six classes of mutations. Mutations from purines on the lagging DNA strand may result from error-prone Translesion Synthesis. **c** The assessment of the length of clustered mutation events on the distances ranging between 2 and 10,000 bp. Sliding window of 5 bp was used to estimate median effect size (black) and its 95% confidence interval (gray) as well as Bonferroni-corrected $-\log 10$ ($P$ value) (red, Wilcoxon signed-rank test, two-sided) for different length of clusters in real data (XP-C leukemia, $n = 6$) against simulations (see "Methods" section). The highest and significant enrichment of clustered mutations was observed for short clusters with lengths between 2 and 16 bp. **d** Intensity of epigenetic marks (5 quantiles) and relative mutation load for XP-C leukemia ($n = 6$, SEMs are indicated). Mutational density on the transcribed, untranscribed DNA strands of genes as well in intergenic regions positively correlated with repressive histone marks H3K27me3 and H3K9me3, and inversely correlated with active chromatin marks (H3K27ac, H3K36me3, H3K4me1). For all three genomic categories, effect of the majority of epigenetic marks was similar. At the same time correlations of untranscribed strand for pyrimidines (or transcribed for purines, red) with H3K27me3 and H34M36me3 were more important than for the other two categories.

into mutations than on the leading strand probably due to more frequent error-prone bypass by translesion synthesis (TLS) polymerases[21,23]. Indeed, we found a strong replicational bias (average 1.38-fold of all six mutational classes in XP-C leukemia, Wilcoxon signed-rank test, two-sided, $P = 2.91e-11$) compatible with preferential bypass of purine DNA lesions by error-prone TLS polymerases on the lagging strand (Fig. 4b and Supplementary Fig. 4c) in XPC-deficient tumors.

TLS polymerases that are recruited to bypass a bulky lesion can also insert incorrect bases opposite to undamaged nucleotides near the lesion[24,25]. Indeed, in all eight XP-C tumors, we observed statistically significant excess of clustered events as compared to the random distribution (Fig. 4c and Supplementary Fig. 5). In diploid genome regions of XP-C leukemia 0.3% of SBS formed 140 short clusters with distance between mutations inferior to 16 bp and mean of 7 bp (Fig. 4c and Supplementary

Fig. 5). Moreover, 6.56-fold more mutations, which occurred within a distance of 16 bp from each other, were colocalized on the same sequencing reads, indicating that clustered mutations affect the same allele and may be interconnected (Wilcoxon signed-rank test, two-sided, $P = 0.031$). These results are compatible with the hypothesis of the existence of bulky DNA lesions that enter the S-phase and get bypassed by error-prone translesion DNA synthesis polymerases[23] in *XPC*-deficient cells, while in *XPC*-proficient cells majority of these lesions may be repaired prior to replication in error-free manner.

Due to the absence of GG-NER we expected to observe strong difference in terms of mutation rates between transcribed and untranscribed strands, particularly in open chromatin and early-replicating regions known to be actively transcribed while we expected no difference between untranscribed strand of genes and intergenic regions in heterochromatic regions[20]. In XP-C leukemia

mutation load in regions of open chromatin was strongly depleted in early-replicating regions and regions with active histone marks (H3K27ac (2.83-fold), H3K36me3 (8.45-fold), H3K4me1 (2.72-fold)) for transcribed strands of genes (Fig. 4a, d). Similar but weaker trends were observed when only untranscribed strands of genes and intergenic regions were analyzed (Fig. 4a, d and Supplementary Fig. 4a). Mutation load was also enriched on the untranscribed strand of genes and intergenic regions with repressive histone marks (H3K27me3 (1.26- and 1.09-fold), H3K9me3 (1.28- and 1.25-fold)) and in late replicating regions associated with heterochromatin (Fig. 4a, d). The observed patterns further confirm effectiveness of TC-NER on transcribed strand of genes in euchromatic regions while prove GG-NER being dysfunctional on both intergenic regions and untranscribed strands of genes all over the genome in XP-C samples. To assess the relative mutation rates in different chromatin state regions, we compared XP-C leukemia samples and sporadic myeloid neoplasms. The analysis revealed more homogeneous mutation load across the different states in XP-C leukemia in comparison with sporadic leukemia as well as elevated mutation rates in heterochromatic regions relative to genic and regulatory elements (Supplementary Fig. 4b).

To further validate the mutational consequences of XPC deficiency, we compared the mutational landscape of cutaneous squamous cell carcinomas (cSCC) from XP-C patients and sporadic tumors[20]. All cSCC tumors, independently of XP-C mutational status presented the typical UV-light induced signature (C > T mutations at YpC sites (where Y designates C or T), 85.6%, Supplementary Fig. 6a), which arises due to the bulky lesions on pyrimidines. However, in XP-C cSCCs there was remarkably more pronounced decrease of mutations from pyrimidines on the transcribed strand relative to untrascribed strand and intergenic regions, as well as much stronger transcriptional bias in highly expressed genes (Supplementary Fig. 6b, c). Moreover, XP-C cSCC demonstrated stronger difference than sporadic cSCC between mutation rate on the transcribed strand of genes on the one side, and untranscribed strand of genes and intergenic regions on the other

(Supplementary Fig. 6b, c, d). These differences were particularly strong in transcriptionally active early-replicating regions (Supplementary Fig. 6d). In the case of XP-C internal tumors the observed patterns were similar with the only difference that the mutational profiles are compatible with mutations from purines (Figs. 3c, f and 4a).

**The majority of mutations precede copy number alterations**. In order to assess the timing of somatic mutations in XP-C tumors, we selected the regions of somatic copy number alterations (SCNAs) where one allele was duplicated. We quantified the number of mutations that occurred before and after SCNA[26] based on variant allele frequencies ($n = 2307$ mutations in four copy neutral LOH and four copy gains; Supplementary Table 2 and Supplementary Fig. 7). On average 75% of mutations occurred before SCNAs suggesting that they may have accumulated in progenitor cells before tumorigenesis or early in tumor development (Wilcoxon signed-rank test, two-sided, $P = 0.03906$; Fig. 5a). Therefore, the observed mutational burden and signature in XP-C tumor genomes may partially represent mutagenesis associated with lesion accumulation during the lifetime of normal body cells (Fig. 5b).

## Discussion

This described mutator phenotype may explain the increased risk of internal cancers in general and particularly for hematological malignancies in XP-C patients, which may be associated with relatively high rate of blood stem cell divisions[27]. Our results are in line with recent reports in human and mice showing that attenuated NER at germinal level is associated with increased risk of lymphoma and sarcoma[28,29].

The derived XP-C cancers Signature "C" has the highest similarity to COSMIC Signature 8, which was originally extracted from sporadic tumors with the most elevated (but not usually exceeding 35%) fraction in sarcoma, medulloblastoma, lymphoma, chronic lymphocytic leukemia and breast cancer[18]. While in some works,

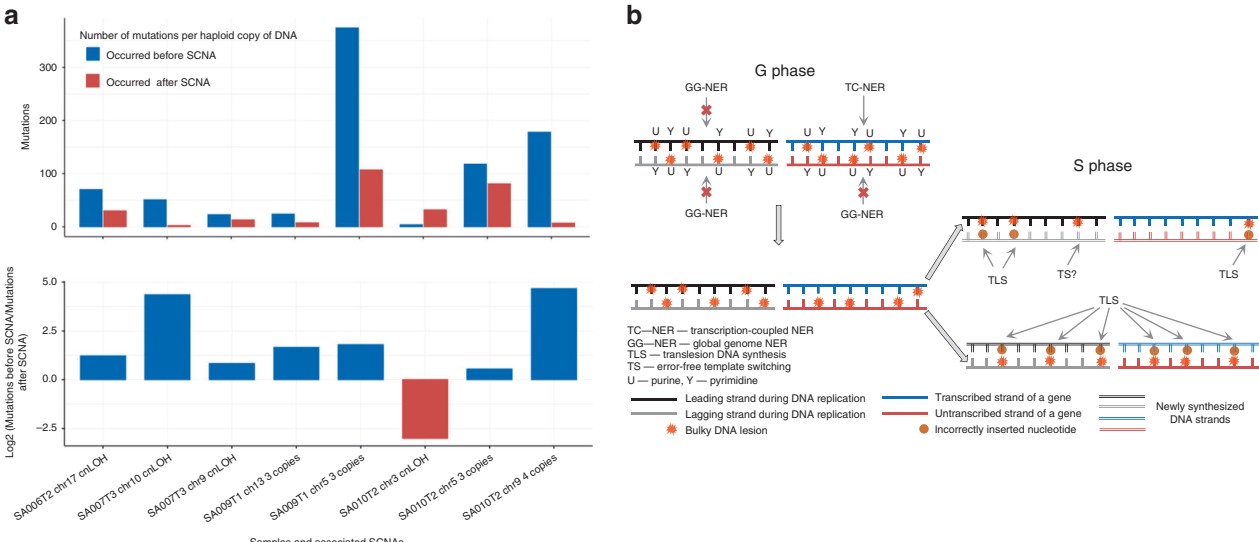

**Fig. 5 Accumulation of DNA lesions and mutations observed in XP-C tumors. a** Relative number of mutations that occurred before and after SCNAs in XP-C cancer genomes (normalized per haploid DNA copy number). The majority of events demonstrate an excess of mutations that were accumulated before the SCNA and may have occurred in tumor-progenitor cells or at early stages of carcinogenesis. **b** A model of DNA lesion accumulation and mutagenesis in XP-C cells. In XP-C cells where GG-NER is dysfunctional, bulky lesions cannot be efficiently repaired and persist everywhere in the genome except transcribed strands of active genes where TC-NER is operative. During the S-phase a part of bulky lesions on the leading strand may be removed by error-free template switching (TS) mechanisms while on the lagging strand they are converted to mutations by error-prone translesion synthesis (TLS) more frequently, causing mutation accumulation with cell divisions and observed transcriptional and replication biases.

it was attributed to homology-repair deficiency[30,31], recently in organoid models Signature 8 was associated with the nucleotide-excision repair deficiency[19]. Comparison of the mutational profiles and NMF-extracted Signature "C" from XP-C internal tumors with the mutational profiles of human *XPC* and mouse *Ercc1* knockouts demonstrated high similarity between them highlighting the dysfunctional NER as the genetic basis of their common mutational process. Our work provides evidence that COSMIC Signature 8 is likely to result from mutagenesis associated with bulky lesions primarily repaired by NER and can be considered as a marker of attenuated NER function.

Taken together our results and previous reports demonstrate that NER deficiency in different tissue types and in in vitro models unmasks a unique mutational process of similar etiology. A broad spectrum of nucleotide substitutions and deletions in XP-C context suggests the existence of a compendium of different bulky lesions induced by one or more genotoxins in DNA of somatic cells. The studied patients were diagnosed as XP-C at early age (median: 3 years) and were well protected from environmental mutagens during their life; therefore, the observed mutagenesis could be caused by endogenous genotoxins which DNA lesions are almost fully repaired in XPC-proficient cells (Fig. 5b).

Future studies on the identification of the nature of this mutational process and its link with particular genotoxins (for example, free radicals, aldehydes, food mutagens) producing bulky lesions may result in the elaboration of preventive measures for XP patients. Except for the breast sarcoma sample from Comorian Archipelago with IVS12 mutation, our dataset mainly represents an XP-C population of the Northern African origin and single *XPC* mutation (delTG) urging the importance of expanding the investigation of internal tumorigenesis and underlying mutagenesis in different XP populations.

## Methods

**Studied samples**. Patients from the study were diagnosed with Xeroderma Pigmentosum at early age (median: 3.5 years; range 1.5−9 years). Primary fibroblasts from sun-unexposed skin were used to determine the DNA repair deficiency by unscheduled DNA synthesis following UV-C irradiation[32]. The XP genetic defect was characterized by complementation assay using recombinant retroviruses expressing wild-type DNA repair genes[33]. The absence of the XPC protein was shown by Western blots[34]. The *XPC* mutation was determined by Sanger sequencing or whole-exome sequencing. Informed signed consents were obtained from patients and/or their parents in accordance with the Declaration of Helsinki and the French law. This study was approved by the French Agency of Biomedicine (Paris, France), by the Ethics Committee from the CPP of Universitary Bordeaux Hospital (Bordeaux, France) and by the Institutional Review Board of the University Institute of Hematology (IUH: Saint-Louis Hospital, Paris). For patients with leukemia ($n = 6$), tumoral bone marrow or peripheral blood mononucleated cells were separated on Fycoll-Hypaque. Cultured skin fibroblast cells were used as non-hematopoietic DNA controls in five out of six patients. In the additional patient, bone marrow CD34+, CD14+ and CD3+ cells were sorted with magnetic beads; CD34+ CD14+ cells represented the leukemic fraction while CD3+ T-lymphocytes, non-leukemic fraction was used as a control. DNA from solid tumors (SA002T2 and SA007T3) was extracted from FFPE blocks after examination and dissection by a pathologist. Tumor DNA was extracted from parts of FFPE containing more than 90% of tumor cells. Germline DNA was extracted from the non-tumoral part of FFPE (Supplementary Table 1).

**Genome sequencing and data processing**. The genomes were sequenced using BGISEQ-500 or Illumina Hiseq 2500 (SA008T6) sequencers according to the manufacturer protocols to the mean coverage after deduplication equal to 45× for tumor and 30× for normal DNA (Supplementary Table 1) using 100 bp paired-end reads. Reads were mapped using BWA-MEM (v0.7.12) software[35] to the GRCh37 human reference genome and then used the standard GATK best practice pipeline[36] to process the samples and call somatic genetic variants. PCR duplicates were removed and base quality score recalibrated using GATK[37] (v4.0.10.1), MarkDuplicates and BaseRecalibrator tools. Somatic SNVs and INDELs were called and filtered using GATK tools Mutect2, FilterMutectCalls and FilterByOrientationBias and annotated with oncotator[38] (v1.9.9.0). SCNAs calling was done with FACETS[39] (v 0.5.14). Quality controls of fastq and mapping were done with FASTQC[40] (v0.11.7), samtools[41] (v1.9), GATK HSmetrics, mosdepth[42] (v0.2.5)

and multiqc[43] (v1.5). All processing steps were combined in a pipeline built with snakemake[44] (v5.4.0).

The cSCC from the work of Zheng et al.[20] were downloaded as SRA files from the database of Genotypes and Phenotypes (dbGaP). The dataset was processed and filtered in the same way as XP-C leukemia samples.

**Filtration of somatic variants**. For XP-C leukemia samples from bone marrow biopsies, we used additional filtration of the PASS variants which included requirement of at least one read on the both strands (F1R2.split (',').1 > 0 && F2R1.split (',').1 > 0 filters in GATK) and the variant allele frequency (VAF) minimal threshold equal to 0.05.

To avoid contamination of true variants by FFPE sequencing artifacts, we used more stringent criteria for breast sarcoma (SA007T3) and rhabdomyosarcoma (SA002T2) samples which included at least 2 and 1 reads from each strand and minimal VAF equal to 0.3 and 0.4 for breast cancer and rhabdomyosarcoma samples respectively. These thresholds were chosen empirically taking into account the high purity/ploidy of the samples (Supplementary Table 1) and VAF of FFPE artefacts which can vary between 0.01 and 0.15[45].

Additionally, all used VCF files were filtered based on the alignability map of human genome[46] from UCSC browser[47] (https://genome.ucsc.edu/cgi-bin/hgFileUi?db=hg19&g=wgEncodeMapability) with the length of K-mer equal to 75 bp (wgEncodeCrgMapabilityAlign75mer, mutations overlapped regions with score <1 were filtered out) and UCSC Browser blacklisted regions (Duke and DAC).

**Mutational signatures analysis**. To convert the VCF files into a catalog of mutational matrices, we used the MutationalPatterns software v.1.11.0[48]. Profiling of the mutational matrices of indels and double nucleotides substitutions was performed with SigProfilerMatrixGenerator v.1.0 software[49].

For comparison with XP-C tumors we used 190 tissue-matched whole cancer genomes from the ICGC PCAWG collection[50] which included cancers from the following projects: Chronic Myeloid Disorders—UK ($n = 57$), Acute Myeloid Leukaemia—KR ($n = 8$), Breast Cancer TCGA US ($n = 91$), Sarcoma—TCGA US ($n = 34$). We used only high-quality variants and additionally filtered out mutations in low-mappability and blacklisted regions of the human genome.

To construct the multidimension scaling plot (MDS), we computed pairwise Cosine similarity distance between all pairs of the samples using MutationalPatterns package[48] and then processed the matrix of distances between the samples in prcomp() function in R.

To perform non-negative Matrix Factorization approach and extract de novo mutational signatures, we used the XP-C samples along with tissue-matched dataset of PCAWG samples ($n = 190$) in NMF framework realized in MutationalPatterns R package[48] with 500 initialization runs. After examination of the diagnostic plots (Supplementary Fig. 2a), we choose $K = 7$ (with RSS at inflation point, according to Hatchins et al.[51]) to extract mutational signatures (Supplementary Fig. 2b) and then assigned them to the known mutational signatures based on the Cosine similarity (Fig. 2c and Supplementary Fig. 2e). Choosing of lower ($K = 4$) or higher factorization rank ($K = 9$) did not influence significantly the extracted Signature "C" and its proportion in samples.

To quantify the contribution of the NMF-derived mutational signatures (A−G) in XP-C tumors and tissue-matched PCAWG cancers, we used the quadratic programming-based algorithm[52] realized in SigsPack R package[53] (Fig. 2b). To better understand and quantify the contribution of the NMF-derived mutational signatures in XP-C dataset, we additionally used bootstrapping ($n = 10,000$) on substitution classes to receive the confidence intervals of each signature contribution (Supplementary Fig. 2d).

**Transcriptional strand bias analysis**. Transcriptional strand bias (TRB) was quantified for each sample and six mutational classes using MutationalPatterns package[48]. The function computed inequality between mutations from pyrimidines (C > A,T,G; T > A,C,G) to mutations from purines (G > A,C,T; A > C,G,T) for genes located on the sense and antisense strands of DNA relative to the reference human genome. Inequality in the number of mutations from purines and pyrimidines was considered as evidence of transcriptional bias and statistical significance was assessed using Poisson test.

To compute tissue-specific TRB between genes expressed at low and high level, we used RPKM values of RNA-seq from Epigenetic Roadmap Project[54] (E028 for breast sarcoma, E050 for leukemia, E100 for rhabdomyosarcoma). For each gene mutations were separated as located on transcribed or untranscribed strands and genes were divided into bins by the level of expression (RPKMs: 0−0.1, 0.1−1, 1−10, 10−20,000 for leukemia; 0−0.1, 0.1−20,000 for breast sarcoma and rhabdomyosarcoma). The significance for each bin was assessed using Poisson test, two-sided (single samples of breast sarcoma and rhabdomyosarcoma) or Wilcoxon signed-rank test, two-sided (leukemia, $n = 6$) and then for visualization the number of mutations was normalized by the total length of genes in each bin.

Following the hypothesis that majority of mutations were caused by purine DNA lesions, we were able to compute strand-specific mutation densities around transcription start sites (TSSs). Transcribed and untranscribed strands of genes as well as 5′ adjacent to TSS intergenic regions were treated separately. TSSs of all

annotated genes (GENECODE v30 [55]) were retrieved using BEDTools v2.29.0 [56] and then regions located ±50 kb of TSSs were split into 1-kb intervals. The 1-kb intervals that overlapped with other intergenic or genic intervals (represented mainly by overlapped or closely located genes) were removed. This approach rendered 237 Mbp of 5′ proximal to TSS intergenic regions and 151 Mbp of genic regions.

**Replication timing**. We used repliseq data from 12 cell lines [57,58] to calculate consensus replication timing regions. For each 1-kb regions we calculated the standard deviation between all the cell lines and removed all regions with standard deviation higher than 15. For the rest of consistent regions across different cell lines, we calculated the mean values and used them during analysis. The genome was divided into five bins ($10-25$, $25-40$, $40-55$, $55-70$, $70-85$) according to the replication timing values and mutational density was calculated for each bin adjusting for the length of each region. We computed dependence of mutational density on replication timing independently for genic and intergenic regions separating mutations on transcribed strand and untranscribed strands.

**Epigenetic marks and mutational density**. To infer relationship between mutation density and intensity of various epigenetic marks (methylation, H3K27ac, H3K27me3, H3K36me3, H3K4me1, H3K9me3), we downloaded bigwig files of the Roadmap Epigenomics Project [54] and converted them to wig and then bed files (tissue E050). The mean intensity of each mark was calculated for 1-kb non-overlapping windows across autosomes with BEDOPS v2.4.37 (bedmap) software [59]. We used only genomic windows with high alignability (equal to 1) along at least 90% of a window. Mark intensities were normalized in $1-100$ range. For each window we split mark intensities into 5 quantiles (cut2() function in R [60]) and calculated the relative mutation density of each mark for intergenic regions, transcribed and untranscibed strands of genes.

The ChromHMM Expanded 18-state models of chromatin states (E050) were downloaded as bed file [54] and all the windows with the highest alignability spanning less than 90% of the window were filtered out. Then we calculated relative mutation density for each sample and chromatin state for XP-C leukemia and sporadic myeloid neoplasms.

**Replicational strand bias**. We used data from Okazaki-seq experiments data [61] for GM06990 and HeLa cell lines to infer the regions of genome preferentially replicating as lagging or leading strand relative to the reference human genome. 1 kbp genomic regions for which values representing the direction of replication fork differed between cell lines >0.4 were removed. We calculated the ratio of the densities between mutations from pyrimidines (C, T) and purines (G, A) for each bin ($-1$ to 0.5, $-0.5$ to 0, 0 to 0.5, 0.5 to 1) of the preferential replication direction (negative values correspond to genomic regions where reference strand is replicated as lagging strand; and positive values—as leading) similar to the methodology of Seplyarskiy et al. [62].

**Clustered mutations**. To evaluate the distribution of mutations across the genome for the presence of clustered mutations in our dataset, we performed Monte Carlo simulations for the intermutation distances distribution of random mutations for ranges between 2 and 10,000 bp for each studied sample. We developed a mathematical model of the Monte Carlo method for random mutations generation based on the following statements: (1) positions of mutations are random and uniformly distributed along the genome; (2) random positions are selected from the same set of genomic intervals as original somatic mutations; (3) the number and nucleotide context spectrum of randomly generated mutations exactly matches somatic mutations in the corresponding sample. As follows, our simulations are based on the discrete homogeneous Poisson point process. The Monte Carlo simulations were performed using Java programming language, discrete random positions were generated with standard Java Random class (Supplementary code). Data analysis was carried out with MathWorks MATLAB. We randomly assigned mutations giving their trinucleotide (3 bp) contexts and repeated the procedure 30,000 times for each sample (Supplementary Fig. 5).

To compute statistics for the distances between neighbors for randomly placed mutations within mappability sections for chromosomes and whole genome, we used the following algorithm:

```
 1:  input: G          ▶ mappable sections of genome
 2:  input: S          ▶ desired statistics of nucleotide contexts
 3:  input: N          ▶ total number of simulations
 4:  input: D          ▶ maximal allowed distance between mutations
 5:  output: M ← {∅}   ▶ empty set for randomly generated mutations
 6:  output: O ← {∅}   ▶ empty set for distance statistics
 7:  repeat N times
 8:     while size of M is less than size of S
 9:        select random position p inside G
10:        determine nucleotide context x for p
11:        if count of x in M is less than in S
12:           append p to M
13:        end if
14:     end while
```

```
15:     sort M
16:     for every position p in M except last
17:        compute distance d between p and next position in M
18:        if d <= D
19:           append d into O
20:        end if
21:     end for
22:     output M
23:     output O
24:  end repeat
```

We next verified that random mutations at small distances produced by random generations followed the Poisson distribution. Then, the means for simulated distributions were compared with the observed intermutation distances for XP-C leukemia samples ($n = 6$) using Wilcoxon signed-rank, two-sided test in 5 bp overlapping (1 bp step) windows to define the length of clusters (for $2-10,000$ bp intervals). Resulted $P$ values were corrected with Bonferroni approach. Significant enrichment of clustered mutations at short distances remained when simulations were performed without taking into account the context of mutations or in 5-bp context of mutations; or when only euploid parts of the genomes were taken into account. Four exomes of XP-C samples were independently sequenced on Illumina Hiseq 2500 with ~100× sequence coverage. Out of six clusters that overlapped exonic regions all six were validated. Additionally, we assessed the number of mutations located on the same read or different reads for clusters up to 16 bp located in diploid genomic regions.

**Relative number of mutations before and after SCNAs**. To infer the relative number of mutations that occurred before and after SCNA, we followed the previously described methodology [26] and identified SCNA of two classes in our dataset: copy gain or cnLOH (Supplementary Table 2). In these SCNA regions taking into account tumor purity and ploidy of the regions, we determined the conservative variant allele frequency (VAF) thresholds to separate the mutations that occurred before and after SCNA given their VAF. The number of mutations was then normalized per haploid copy of a genomic segment.

**Reporting summary**. Further information on research design is available in the Nature Research Reporting Summary linked to this article.

## Data availability

Experimental data generated in this study have been deposited to the European Genome-Phenome Archive (EGA), the accession number is EGAS00001004511. The PCAWG data referenced in the study (consensus VCF files with SNVs and INDELs) are available in a public repository from the https://dcc.icgc.org/repositories website. Genomic dataset of cSCC used in this study is available in the dbGaP database under accession code phs000830.v1.p1. All the other data supporting the findings of this study are available within the article and its supplementary information files and from the corresponding author upon reasonable request. A reporting summary for this article is available as a Supplementary Information file.

## Code availability

All software used is published and/or in the public domain. Custom Java code for the clustered mutation analysis is available as the Supplementary code.

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

## Acknowledgements

S.N. was supported by grant Foundation ARC 2017, Foundation Gustave Roussy and Swiss Cancer League KFC-3985-08-2016. B.T.M. was supported by MES RK grant AP05134722 and NU ORAU grant 091019CRP2111. The authors would like to thank Dr. Patricia Kannouche and Dr. V.B. Seplyarskiy for fruitful discussions and participation, and Dr. F. Rajabi, Dr. Catherine Genestie and Dr. Samuel Quentin for DNA extraction and providing samples. The authors are also very thankful to Dr. C. Genestie (IGR, Villejuif, France), Dr. Z. Tata and Dr. S. Duquenne and Dr. F. Cartault for giving us or for manipulating biopsies of tumors and Xiaole Xu (BGI) for the management of sequencing.

## Author contributions

S.N., A.S. and A.A.Y. designed the study. A.S. and J.S. collected the samples. A.A.Y. performed the data analysis and prepared figures. B.T.M. participated in the data analysis. I.P. performed data preprocessing. A.A.Y. and S.N. drafted manuscript. A.S. and J.S. commented manuscript. All authors contributed to the final version of the paper.

## Competing interests

The authors declare no competing interests.
