## [Peer Review File · Nature Communications]

Response to reviewers:

REVIEWERS' COMMENTS:

Reviewer #1 (Remarks to the Author):

The authors have adequately addressed this reviewer's concerns.

Reply: we thank the reviewer for the valuable comments which helped to improve our manuscript.

Reviewer #3 (Remarks to the Author):

The authors have adequately addressed my concerns. However although their prior MS (Sarasin Blood 2019 PMID 30914417) does report somatic mutations, these important details, and comparison with sporadic tumors remain absent from the MS. This should be corrected and should be a simple fix.

Reply: we now included comparison of the most common somatic aberrations in our samples with TCGA AML dataset (L102-111). We thank the reviewer for the valuable comments which helped to improve our manuscript.

Reviewer #5 (Remarks to the Author):

My comments and requests for additional supplementary information have been satisfactorily addressed, and, in my view, the other reviewers' comments have also been satisfactorily addressed.

Reply: we appreciate the encouraging summary of our revision and thank the reviewer for the valuable comments which helped to improve our manuscript.

I note a citation problem on lines 703 and 843, where the author is listed as "Consortium".

Reply: we corrected bibliography.

Reviewer #1 (Remarks to the Author): Expert in DNA damage

Title: XPC deficiency increases risk of hematologic malignancies through mutator phenotype and specific mutational signature

Andrey A. Yurchenko, Ismael Padioleau, Bakhyt T. Matkarimov, Jean Soulier, Alain Sarasin, Sergey Nikolaev

Summary: XPC-deficiency reduces the ability of global genome NER (GG-NER) to locate and subsequently repair bulky DNA lesions. This condition increases the risk of developing tumors and is often associated with skin tumors, but individuals are also more at risk for developing internal tumors. Here, the authors analyze internal tumor samples from XPC-deficient and non-XPC-deficient individuals. They use a previously established approach to detect mutational signatures that are unique to the XPC-deficient samples, and identify patterns within those signatures. The XPC-deficient samples have a mutational profile that clusters separately from the control samples. They also found a higher proportion of mutated purines on untranscribed DNA strands, intergenic regions, and silent genes than purines on transcribed DNA strands of expressed genes. Lastly, they find that most of the mutations occurred prior to somatic copy number alterations. The authors conclude the XPC-deficiency results in un-repaired bulky lesions, which translesion synthesis often bypasses in a mutagenic fashion during replication.

While this is an interesting and timely study, the impact could be strengthened by providing some mechanistic insight to support and solidify their conclusions.

Comments/Concerns:

1. The authors do provide novel insight into mutagenic patterns associated with XPC-deficiency in internal cancers. They reveal transcriptional bias in multiple tumor types at a genomic level using sophisticated analyses. However, they primarily provide evidence of a potential mechanism rather than testing the proposed mechanism itself. Additionally, there are positive controls are lacking (for example, UV-damaged XPC-deficient skin cancer samples) from the analyses, which would strengthen their experimental design.

2. The authors do not discuss why pyrimidines on untranscribed strands would be repaired (but purines on untranscribed strands would) even if GG-NER was not functional due to XPC-deficiency. Discussion of the differences between pyrimidines and purines with respect to transcriptional bias is warranted. Related to this, in lines 137 to 139 the authors mention that the pattern they see could be the result of excess pyrimidine mutations or reduced purine mutations, but they do not discuss this in detail, and they do not provide evidence to support their comments. They should provide additional data to support a potential explanation.

3. Some of the figure legends and axes could be labeled more clearly: In Fig. 1C it could be made clearer that the lower axis represents the nucleotides upstream and downstream of the nucleotide in the top panel; In Fig. 2B the figure legend (A, B, C, D, E, F, G) could be labeled (Signature "A", Signature "B", etc.) to make it more clear what the letters are representing; When labeling the red and the blue lines for Fig. 3, if purines are the focus (there are excess purine mutations on the untranscribed strand due to XPC deficiency) then purines should be listed first (e.g. red is transcribed for purines, untranscribed for pyrimidines) rather than pyrimidines listed first; Figure 3F should specify if pyrimidines or purines are the nucleotides on the untranscribed or transcribed strand.

4. The description in Fig. 4A states that the pattern seen is due to the absence of GG-NER. However, this is an assumption and would be more appropriate in the discussion section than a figure description, and/or supported by additional data.

5. There are several small typos that should be corrected: line 50 - "untranscribed strands"; line 153 - "untranscribed strands"; line 155 - "strands"; 158 - "damage is on purine bases", 176 - "within a distance", 179 - "of the existence"

Reviewer #3 (Remarks to the Author): Expert in leukemia genomics

This is an interesting study that suggests that the genetic defect in XPC promotes a singular mutational signature with strand bias, which has potential mechanistic implications. I would suggest the following:

It would be more compelling that the signature is pathogenic if the underlying mutated gene(s) are perturbed in an experimental model and this is sequenced to show that this perturbation induces the same signature.

While the signature may be characteristic, it is not a "specific" signature as the title is written to suggest - it is an established COSMIC signature seen in other contexts. While it may be that bulky nucleotides may be responsible, it would be helpful if the authors could explore/explain WHY this specific pattern of mutated residues is observed.

The correlation with epigenetic state is potentially interesting but underdeveloped as correlation has only been performed with individual marks. It would be helpful if the authors correlate with chromatin state analysis that combines marks (e.g. ChromHMM).

The genomic description is focused on mutation burden and signature but is otherwise limited - what driver genes or others are mutated by sequence or structural variation? This could and should be examined and presented in more detail (and compared to sporadic tumors).

The actual name of the causal gene (Table 1, and in the text) should be stated.

Reviewer #5 (Remarks to the Author): Expert in DNA damage and genomics

The manuscript by Yurchenko and colleagues is a study of somatic mutations in 6 leukemias and 2 sarcomas in xeroderma pigmentosum group C (XP-C) patients. Since the absent protein, XPC, is important for global excision repair, we would expect that non-UV related tumors in the patients will have distinctive mutational patterns. Indeed, this paper reports elevated mutation rates and a distinctive pattern of somatic single base mutations (single base substitutions, SBSs) in these tumors. The paper also reports *extremely* strong transcriptional strand asymmetry, which is consistent with the operation of transcription coupled nucleotide excision repair in the absence of XPC-mediated global excision repair. The paper also substantiates a pattern of mutations consistent with elevated error-prone translesion synthesis on the lagging versus leading replication strand opposite purines with bulky adducts.

I believe the conclusions of the paper are substantially correct, and the analysis is both careful and thorough. Some notable strengths include the analysis of interaction of the mutation density with transcriptional and replication strand bias. The extremely strong transcriptional strand asymmetry is important confirmatory evidence that this signature is free of sequencing artifacts and distinct from SBS8.

This paper constitutes an important advance in the study of the function of XPC as reflected by the consequences of its absence.

I have no concerns about the statistical analyses or bioinformatic analyses except for a request to provide the code for the analysis of clustered mutations, as noted below.

ESSENTIAL CHANGES NEEDED

The authors absolutely must provide the final lists of filtered variant calls either as supplementary information, or, if these are considered protected information, on an appropriate archive.

The authors absolutely must provide the single base substitution (SBS), doublet base substitution

(DBS) and indel spectra in numerical form (i.e. in Excel or .CSV files or as VCFs).

Presence of the data in EGA is essential. I would encourage the authors to submit immediately (the data can be embargoed). Getting data uploaded and released on EGA can be slow.

OTHER MAJOR COMMENTS

It is essentially impossible to give a precise verbal description of the analysis of clustered mutations (lines 608 – 637). I strongly suggest providing the code.

The double base substitution and indel substitution patterns are also quite distinctive. This is worth a mention in the results.

PRESENTATION COMMENTS

The paper is quite dense, with many long and slightly damaged sentences. The authors would do well to follow the advice I give trainees in my lab: If English is not your mother tongue, do not write long sentences. Actually, even if it is your mother tongue, do not write long sentences. Why make it harder for others to understand your work?

Some specific suggestions on presentation:

The point made on line 168 through 172 is quite interesting but rather buried. Suggest starting a new paragraph at "TLS polymerases which are recruited..."

At the beginning of results suggest that you clarify that you sequenced cancer and non-cancer to identify cancer-specific somatic mutations.

MINOR TYPOS, GRAMMAR, PRESENTATION ISSUES (NOT EXHAUSTIVE):

Line 25 – "internal cancers" – Suggest "non-skin cancers" would be clearer.

Lines 28-29 "conferring to its elevated incidence and early appearance". I think the intent is ", which we presume leads to the elevated incidence and early appearance of leukemias in these patients"

Line 42 not sure what "and XP variant" refers to.

Line 44 "on" -> "into"

Line 65, delete "the" in "the age"

Lines 237 – 239 seem out of sync with the paper, as the Comoros population is not genetically North African and the patient had the Comorian IVS12 variant.

Line 627 "tow" -> "two"

SIGNATURE

Steven G. Rozen

REVIEWER COMMENTS

Reviewer #1 (Remarks to the Author): Expert in DNA damage

Title: XPC deficiency increases risk of hematologic malignancies through mutator phenotype and specific mutational signature

Andrey A. Yurchenko, Ismael Padioleau, Bakhyt T. Matkarimov, Jean Soulier, Alain Sarasin, Sergey Nikolaev

Summary: XPC-deficiency reduces the ability of global genome NER (GG-NER) to locate and subsequently repair bulky DNA lesions. This condition increases the risk of developing tumors and is often associated with skin tumors, but individuals are also more at risk for developing internal tumors. Here, the authors analyze internal tumor samples from XPC-deficient and non-XPC-deficient individuals. They use a previously established approach to detect mutational signatures that are unique to the XPC-deficient samples, and identify patterns within those signatures. The XPC-deficient samples have a mutational profile that clusters separately from the control samples. They also found a higher proportion of mutated purines on untranscribed DNA strands, intergenic regions, and silent genes than purines on transcribed DNA strands of expressed genes. Lastly, they find that most of the mutations occurred prior to somatic copy number alterations. The authors conclude the XPC-deficiency results in un-repaired bulky lesions, which translesion synthesis often bypasses in a mutagenic fashion during replication.

While this is an interesting and timely study, the impact could be strengthened by providing some mechanistic insight to support and solidify their conclusions.

Comments/Concerns:

1. The authors do provide novel insight into mutagenic patterns associated with XPC-deficiency in internal cancers. They reveal transcriptional bias in multiple tumor types

at a genomic level using sophisticated analyses. However, they primarily provide evidence of a potential mechanism rather than testing the proposed mechanism itself. Additionally, there are positive controls are lacking (for example, UV-damaged XPC-deficient skin cancer samples) from the analyses, which would strengthen their experimental design.

Response: We thank the reviewer for proposing a relevant validation experiment using publicly available data on XP-C. Indeed, the objective of this study was a description of mutational profiles in internal tumors in the context of XPC deficiency where we observe striking differences with WT malignancies including very strong transcriptional bias and repair deficiency on the untranscribed strand of the genes and in intergenic regions. Following recommendation of the reviewer we performed additional analysis using previously published WT and XP-C cutaneous squamous cell carcinoma samples (cSCC) from Zheng et al. 2014 (PMID: 25456125). Now we added this data as the Supplementary Figure 6 and also to Results (L225-241) and Methods (L593-596). In these samples a typical UV-light induced signature (C>T mutations at Py-Py sites, 85.6%) is predominant. When we compared the XP-C cSCC and WT cSCC we observed substantially more pronounced difference between mutation rate on transcribed strand of genes on the one side and untranscribed strand of genes and intergenic regions on the other. Wealth of experimental data established that mutagenesis in skin cancer is associated with UV-induced pyrimidine lesions. We clearly observe that in XP-C cSCC there is remarkably more pronounced decrease of mutations from pyrimidines on the transcribed strand relative to untranscribed strand and intergenic regions (Supplementary Figure 6b,c). In the case of XP-C internal tumors the observed patterns are similar with the only difference that the mutational profiles are compatible with mutations from purines (Figure 3c, f). As the reviewer correctly noticed, to validate the proposed mechanism, a bunch of in vitro experiments testing the mutagenic consequences of various candidate genotoxins would be required, however it was outside the scopes of this study. Moreover, the mutational profiles in studied tumors are characterized by mutations in broad range of contexts (“flat” profile) which may suggest contribution of a set of different genotoxins.

2. The authors do not discuss why pyrimidines on untranscribed strands would be repaired (but purines on untranscribed strands would) even if GG-NER was not functional due to XPC-deficiency. Discussion of the differences between pyrimidines and purines with respect to transcriptional bias is warranted. Related to this, in lines 137 to 139 the authors mention that the pattern they see could be the result of excess pyrimidine mutations or reduced purine mutations, but they do not discuss this in detail, and they do not provide evidence to support their comments. They should provide additional data to support a potential explanation.

Response: Our approach is agnostic, and is not based on a prior hypothesis, if the mutations occur due to lesions at purine or pyrimidine bases. Tumor DNA sequence with the mutations contains changes of both purine and pyrimidine nucleotides. For example, in skin cancer, where a predominant mutational process is associated with mutations of cytosine to thymine (C>T), G>A changes will represent cytosine mutations on the complementary strand. Likewise, in our case we considered that the reverse complement of each mutation represents the same mutation but on a complementary strand; and comparison of such mutations on the transcribed and untranscribed strands of genes revealed a strong bias. This observed bias could be caused by decrease of mutations (from purines) or increase of mutations (from pyrimidines) on the transcribed strand. Both excess and depletion of mutation rates on the transcribed strand are described in the literature (see Haradhvala et al. 2016, Cell, PMID: 26806129) and are referred to Transcription-coupled repair (TCR) or Transcription-coupled Damage (TCD). In order to discriminate between these two possibilities a comparison between mutation rates in intergenic and genic regions can be performed. In case of TCD the increase of mutation rates in gene as compared to intergenic region should be observed (TCD in liver cancer analysis in Haradhvala et al. 2016), while in case of TCR, it should be a decrease of mutation rates in gene as compared to intergenic regions (an example of UV mutational profiles in skin cancer). In our case we observe no difference between mutation rates in pyrimidines on transcribed strand (or purines on untranscribed) in genic and intergenic regions, but we observe a strong decrease of mutations in purines on the transcribed strand (or

pyrimidines on untranscribed) in genes as compared to intergenic regions (Figure 3f). This analysis is compatible with the decrease of mutations from purines on the transcribed strand resulting from the activity of TC-NER. Decrease of mutation rates on the untranscribed strand as compared to transcribed strand or intergenic regions is highly unlikely, and to our knowledge was not described in the literature, therefore we do not pursue this possibility. See also the response to the previous comment where symmetrical situation for UV-induced mutations from pyrimidines in XP-C skin cancer is discussed. We added to the Results section an explanation of patterns in XP-C cSCC in comparison with XP-C internal cancers (L148-156 and L231-241).

3. Some of the figure legends and axes could be labeled more clearly: In Fig. 1C it could be made clearer that the lower axis represents the nucleotides upstream and downstream of the nucleotide in the top panel; In Fig. 2B the figure legend (A, B, C, D, E, F, G) could be labeled (Signature “A”, Signature “B”, etc.) to make it more clear what the letters are representing; When labeling the red and the blue lines for Fig. 3, if purines are the focus (there are excess purine mutations on the untranscribed strand due to XPC deficiency) then purines should be listed first (e.g. red is transcribed for purines, untranscribed for pyrimidines) rather than pyrimidines listed first; Figure 3F should specify if pyrimidines or purines are the nucleotides on the untranscribed or transcribed strand.

Response: We used conventional representation of mutations in trinucleotide context and now added a clarification regarding the X-axis (L339). We changed the letters (A,C,B...) to Signature A,C,B... (Figure 2b). We changed the labels on the Figures 3 and 4 as was suggested by the reviewer putting changes from purine bases first. Figure 3f labelling was clarified in respect of transcribed/untranscribed strands and types of nucleotides.

4. The description in Fig. 4A states that the pattern seen is due to the absence of GG-

NER. However, this is an assumption and would be more appropriate in the discussion section than a figure description, and/or supported by additional data.

Response: We agree with the reviewer, and modified the legend using more accurate wording:” .. which is compatible with the absence of GG-NER” (L391)

In this analysis we hypothesized that due to the absence of GG-NER we should observe strong difference in terms of mutation rates between transcribed and untranscribed strands, particularly in early replicating regions known to be actively transcribed while we expect no difference between untranscribed strand of genes and intergenic regions at any genomic region. Indeed, expected mutation patterns were observed in XP-C internal cancers as well as in XP-C cSCC (the additional analysis), but were less pronounced in tissue-matched sporadic tumors (Supplementary Figure 4a, 6d; Figure 4a).

Now we included the rationale for this assumption into the Results section (L202-206) and compared the differences seen in terms of replicational timing and intergenic, genic transcribed, and genic untranscribed regions between WT and XP-C internal cancers as well as skin cancers with known pattern of UV-induced DNA damage (L234-241).

5. There are several small typos that should be corrected: line 50 - “untranscribed strands”; line 153 - “untranscribed strands”; line 155 - “strands”; 158 - “damage is on purine bases”, 176 - “within a distance”, 179 - “of the existence”

Response: Corrected.

Reviewer #3 (Remarks to the Author): Expert in leukemia genomics

This is an interesting study that suggests that the genetic defect in XPC promotes a singular mutational signature with strand bias, which has potential mechanistic implications. I would suggest the following:

1. It would be more compelling that the signature is pathogenic if the underlying mutated gene(s) are perturbed in an experimental model and this is sequenced to show that this perturbation induces the same signature.

Response: We thank the reviewer for pointing this out that experimental validation with GG-NER/XPC knockout would strengthen the message of the paper. As matter of fact such experiment has already been performed by the team of Edwin Cuppen (Jager et al. 2019, Genome Research, PMID: 31221724). Specifically, they tested mutational consequences of NER deficiency in mouse liver (*Ercc1*) as well as in human intestinal organoid culture (*XPC*). Strikingly in both systems a signature similar to COSMIC Signature 8 was enriched in NER-deficient context but not in the wild-type control. In the original version of our paper we compared mutational signatures from XP-C internal cancers with the *Ercc1* and *XPC* knockouts from Jager et al. 2019 and revealed high similarity between all of them and also Signature "C" (Figure 2c). At the same time, we probably did not discuss sufficiently the results of this analysis. We agree with the reviewer that this piece of information solidify the results and conclusions from this study and added corresponding text to Discussion (L269-273).

2. While the signature may be characteristic, it is not a "specific" signature as the title is written to suggest - it is an established COSMIC signature seen in other contexts. While it may be that bulky nucleotides may be responsible, it would be helpful if the authors could explore/explain WHY this specific pattern of mutated residues is observed.

Response: The signature “C” had moderate Cosine similarity (0.86) with the Signature 8 from the pancancer analysis (COSMIC) being different by specific trinucleotide contexts (VpCpT > D and NpCpT > T where V designates A,C,T and D – A,G,T; Figure 2a) and more importantly had much more pronounced transcriptional bias thereafter we preferred to describe exactly the similarities and differences between the signatures instead of designating this signature (“C”) to Signature 8. Reviewer 5 also noticed the important differences between the Signature “C” and Signature “8”. We corrected the title changing the word “specific” to “characteristic”.

3. The correlation with epigenetic state is potentially interesting but underdeveloped as correlation has only been performed with individual marks. It would be helpful if the authors correlate with chromatin state analysis that combines marks (e.g. ChromHMM).

Response: Following this valuable suggestion, we performed an additional analysis using 18-state ChromHMM for XP-C and sporadic leukaemia samples. The text describing the analysis is added to Methods (L702-706), Results (L218-224) and as Supplementary Figure 4b.

4. The genomic description is focused on mutation burden and signature but is otherwise limited - what driver genes or others are mutated by sequence or structural variation? This could and should be examined and presented in more detail (and compared to sporadic tumors).

Response: We agree with the reviewer that this is an important piece of information but this question has already been discussed in the recent paper by us which described for the first time the epidemiologic phenomenon of significantly elevated risk of leukemia in XP patients (Sarasin et al. 2019, Blood, PMID: 30914417).

5. The actual name of the causal gene (Table 1, and in the text) should be stated.

Response: In the header of the Table 1 we mentioned that the cause of XPC deficiency is the *XPC*^{-/-} while the list of drivers for these leukemia patients is reported in the Supplementary Table 1 and in Sarasin et al. 2019.

Reviewer #5 (Remarks to the Author): Expert in DNA damage and genomics

The manuscript by Yurchenko and colleagues is a study of somatic mutations in 6 leukemias and 2 sarcomas in xeroderma pigmentosum group C (XP-C) patients. Since the absent protein, XPC, is important for global excision repair, we would expect that non-UV related tumors in the patients will have distinctive mutational patterns.

Indeed, this paper reports elevated mutation rates and a distinctive pattern of somatic single base mutations (single base substitutions, SBSs) in these tumors. The paper also reports **extremely** strong transcriptional strand asymmetry, which is consistent with the operation of transcription coupled nucleotide excision repair in the absence of XPC-mediated global excision repair. The paper also substantiates a pattern of mutations consistent with elevated error-prone translesion synthesis on the lagging versus leading replication strand opposite purines with bulky adducts.

I believe the conclusions of the paper are substantially correct, and the analysis is both careful and thorough. Some notable strengths include the analysis of interaction of the mutation density with transcriptional and replication strand bias. The extremely strong transcriptional strand asymmetry is important confirmatory evidence that this signature is free of sequencing artifacts and distinct from SBS8.

This paper constitutes an important advance in the study of the function of XPC as reflected by the consequences of its absence.

I have no concerns about the statistical analyses or bioinformatic analyses except for a request to provide the code for the analysis of clustered mutations, as noted below.

Response: We thank the reviewer for this encouraging summary of our manuscript.

ESSENTIAL CHANGES NEEDED

1. The authors absolutely must provide the final lists of filtered variant calls either as supplementary information, or, if these are considered protected information, on an appropriate archive.

Response: We are in the process of the uploading raw data (FASTQ) and filtered variants (VCF files) to the EGA therefore it should not delay the publication.

2. The authors absolutely must provide the single base substitution (SBS), doublet base substitution (DBS) and indel spectra in numerical form (i.e. in Excel or .CSV files or as VCFs).

Response: We added the Supplementary Tables 2,3,4 which contain the SBS, DBS and ID spectra.

3. Presence of the data in EGA is essential. I would encourage the authors to submit immediately (the data can be embargoed). Getting data uploaded and released on EGA can be slow.

Response: We thank the reviewer for this advice and have already started to upload the data.

OTHER MAJOR COMMENTS

4. It is essentially impossible to give a precise verbal description of the analysis of clustered mutations (lines 608 – 637). I strongly suggest providing the code.

Response: We provide the code as an additional supplementary file (Supplementary code) and the pseudocode for the analysis is provided now as a part of Methods

(L735-762)

5. The double base substitution and indel substitution patterns are also quite distinctive. This is worth a mention in the results.

Response: We added the description of indels and double base substitutions to the Results (L108-112).

PRESENTATION COMMENTS

6. The paper is quite dense, with many long and slightly damaged sentences. The authors would do well to follow the advice I give trainees in my lab: If English is not your mother tongue, do not write long sentences. Actually, even if it is your mother tongue, do not write long sentences. Why make it harder for others to understand your work?

Response: We highly appreciate the constructive suggestion regarding the long sentences and will apply this principle in the future. However, we tried to cut some long sentences where possible.

Some specific suggestions on presentation:

7. The point made on line 168 through 172 is quite interesting but rather buried. Suggest starting a new paragraph at "TLS polymerases which are recruited..."

Response: We changed the text accordingly (L187).

8. At the beginning of results suggest that you clarify that you sequenced cancer and non-cancer to identify cancer-specific somatic mutations.

Response: We added the clarification (L85).

MINOR TYPOS, GRAMMAR, PRESENTATION ISSUES (NOT EXHAUSTIVE):

9. Line 25 – "internal cancers" – Suggest "non-skin cancers" would be clearer.

Response: Corrected (L25)

10. Lines 28-29 "conferring to its elevated incidence and early appearance". I think the intent is ", which we presume leads to the elevated incidence and early appearance of leukemias in these patients"

Response: Corrected (L29)

11. Line 42 not sure what "and XP variant" refers to.

Response: This is a traditional designation of XP subgroups which can be not very intuitive: XP-A/B/C/D/E/F/G with the loss of NER proteins named as “groups” and XP-V with the loss of polymerase η named as a “variant”. Thus, we decided to leave as it is.

Line 44 "on" -> "into"

Response: Corrected (L45)

Line 65, delete "the" in "the age"

Response: Deleted.

Lines 237 – 239 seem out of sync with the paper, as the Comoros population is not genetically North African and the patient had the Comorian IVS12 variant.

Response: We improved the sentence (L289-290)

Line 627 "tow" -> "two"

Response: Corrected.

REVIEWERS' COMMENTS:

Reviewer #1 (Remarks to the Author):

The authors have adequately addressed this reviewer's concerns.

Reviewer #3 (Remarks to the Author):

The authors have adequately addressed my concerns. However although their prior MS (Sarasin Blood 2019 PMID 30914417) does report somatic mutations, these important details, and comparison with sporadic tumors remain absent from the MS. This should be corrected and should be a simple fix.

Reviewer #5 (Remarks to the Author):

My comments and requests for additional supplementary information have been satisfactorily addressed, and, in my view, the other reviewers' comments have also been satisfactorily addressed.

I note a citation problem on lines 703 and 843, where the author is listed as "Consortium".

Signature

Steven G. Rozen